# A Comprehensive Dictionary and Term Variation Analysis for COVID-19 and SARS-CoV-2

**Robert Leaman and Zhiyong Lu**

National Center for Biotechnology Information (NCBI)
National Library of Medicine (NLM), National Institutes of Health (NIH)
Bethesda, Maryland, USA
`robert.leaman@nih.gov, zhiyong.lu@nih.gov`

## Abstract

The number of unique terms in the scientific literature used to refer to either SARS-CoV-2 or COVID-19 is remarkably large and has continued to increase rapidly despite well-established standardized terms. This high degree of term variation makes high recall identification of these important entities difficult. In this manuscript we present an extensive dictionary of terms used in the literature to refer to SARS-CoV-2 and COVID-19. We use a rule-based approach to iteratively generate new term variants, then locate these variants in a large text corpus. We compare our dictionary to an extensive collection of terminological resources, demonstrating that our resource provides a substantial number of additional terms. We use our dictionary to analyze the usage of SARS-CoV-2 and COVID-19 terms over time and show that the number of unique terms continues to grow rapidly. Our dictionary is freely available at https://github.com/ncbi-nlp/CovidTermVar.

## 1 Introduction

In a public health crisis involving novel entities, such as COVID-19, the disease caused by the SARS-CoV-2 virus, new scientific insights must be disseminated rapidly. Researchers must therefore refer to the novel entities before many of their properties are understood and appropriate terminology can be proposed. For example, the first reports of the condition now known as COVID-19 only refer to a general description of symptoms, such as "pneumonia of unknown aetiology" (Bogoch et al., 2020). The World Health Organization proposed the provisional name "2019-nCoV" for the new virus on January 30, 2020 (World Health Organization, 2020a),

prompting additional terms for the disease, including "2019-nCoV infection" (Huang et al., 2020). The virus was then officially named "SARS-CoV-2" on February 11, 2020 (Coronaviridae Study Group of the International Committee on Taxonomy of Viruses, 2020; Gorbalenya, 2020), with the disease officially named "COVID-19" in a separate proposal on the same day (World Health Organization, 2020b).

Despite these proposals for standardized terminology, the number of unique terms used in the literature to refer to either COVID-19 or SARS-CoV-2 has increased rapidly. Figure 1 visualizes an illustrative sample of terms for COVID-19 and SARS-CoV-2 as a timeline of their first use. Some authors may be dissatisfied with the term "SARS-CoV-2" due to the potential for confusion with "SARS-CoV" (Jiang et al., 2020). However, terminology that does not follow the existing nomenclature standards is common in the biomedical literature (Cohen & Demner-Fushman, 2014), and it seems likely that the exceptionally rapid growth of the literature discussing SARS-CoV-2 and COVID-19 (Teixeira da Silva, 2020) would exaggerate this effect. Handling term variation is critical for automatically identifying mentions of specific entities in text (Leaman et al., 2020), thus despite many mentions of COVID-19 and SARS-CoV-2 following the standardized terminology, identifying these important entities comprehensively – that is, with high recall – requires this high degree of variation to be addressed.

While there have been many efforts to provide extensive resources related to COVID-19 (Chen et al., 2020; Lu Wang et al., 2020) and systems that address identification of SARS-CoV-2 and COVID-19 (Colic et al., 2020; Wang et al., 2020), there have been few efforts to provide comprehensive resources of terms for COVID-19 or SARS-CoV-2 in the scientific literature. Many

Figure 1. Sample of terms used in the literature to refer COVID-19 (top) and SARS-CoV-2 (bottom), illustrating the high degree of terminology variation. Terms are visualized as a timeline of the week of their first appearance in the literature; for example, week 3 (leftmost cell) began January 13, 2020.

| Week | COVID-19 (top) | SARS-CoV-2 (bottom) |
|---|---|---|
| 3 | pneumonia of unknown aetiology | 2019-nCov novel coronavirus |
| 4 | 2019 nCov infection | Wuhan coronavirus |
| 5 | novel coronavirus pneumonia | new coronavirus |
| 6 | COVID-19 SARS-CoV-2 infection | WH-Human 1' coronavirus |
| 7 | coronavirus disease 2019 | SARS-CoV-2 |
| 8 | COVID | severe acute respiratory syndrome coronavirus 2 |
| 9 | 2019 novel coronavirus infection disease | COVID-19 coronavirus |
| 10 | nCOVID-19 | SARS coronavirus 2 |
| 11 | severe acute respiratory syndrome coronavirus 2 infection | new CoV |
| 12 | Wuhan coronavirus pneumonia | COVI-19 |
| 13 | Coronavirus disease of 2019 | SARS COVID 2 |
| 14 | CoV 19 infection | New Cov 19 |
| 15 | COIVD-19 disease | HCoV-19 |
| 16 | COVID-19 ARDS nCov-19 infection | COVID-19 CoV |
| 17 | SARS-CoV-2 infectious disease CV-19 | novel acute respiratory syndrome coronavirus 2 |
| 18 | Novel COVID-19 | NCoV-19 SARS-nCoV-2 |
| 19 | coronavirus 2 syndrome | |
| 20 | SARS-CoV-2 associated ARDS | SC2 |

terminological resources only include a few common variants. For example, Medical Subject Headings (MeSH) lists 12 terms for COVID-19[1] while the NCBI Taxonomy lists 12 terms for SARS-CoV-2[2]. A notable exception is Rashed et al. (2020), who provide extensive dictionaries for both COVID-19 and SARS-CoV-2, though the last update to the dataset was in April 2020[3].

## 2 Methods

We approach the task of creating a dictionary of terms for the virus SARS-CoV-2 and the disease COVID-19 by identifying terms that refer to these in the biomedical literature. While this approach has some similarities to named entity recognition (NER), our task differs from NER in at least two ways. First, NER is typically concerned with a class of similar entities, such as diseases, while our task is only concerned with two specific entities. Second, while named entity recognition must consider each appearance of a potential term in context, in our task we only need consider whether the potential term is used to refer to the entity at least once.

### 2.1 Terminology Guidelines

In this section we provide a high-level description of the guidelines we used to create the dictionary.

- The SARS-CoV-2 virus and the disease it causes (COVID-19) are separate entities.
- Subtypes (e.g. "mild") are not differentiated.
- We do not include "pandemic" as part of the name for either SARS-CoV-2 or COVID-19.
- Terms are associated with either SARS-CoV-2 or COVID-19, but typically not both.
- Ambiguous terms are allowed if their usage in context is not ambiguous.

### 2.2 Description of our approach

Our base lexicon contains 35 terms for COVID-19, including "coronavirus disease 2019" and "nCOVID-2019," and 66 terms for SARS-CoV-2, including "severe acute respiratory syndrome coronavirus 2019" and "hCoV 2019."

Our system expands the set of base terms into a large set of potential terms using three types of rules. The first type of rule substitutes synonyms, such as "new" for "novel," near synonyms, such as "19" for "2019," or hypernyms such as "virus" for

---

[1] https://meshb.nlm.nih.gov/record/ui?ui=C000657245

[2] https://www.ncbi.nlm.nih.gov/Taxonomy/Browser/wwwtax.cgi?id=2697049

[3] https://github.com/Aitslab/corona

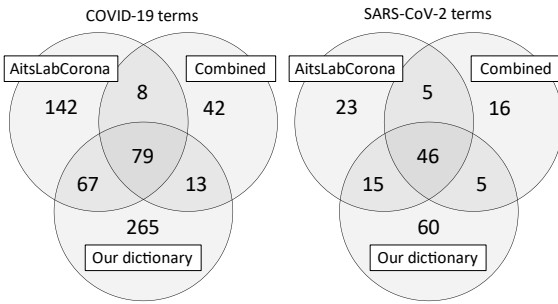

Figure 2. Number of unique terms in common between the *AitsLabCorona* dictionary, *Combined* dictionary and our dictionary for both COVID-19 and SARS-CoV-2.

"coronavirus." The second rule type expands a term for COVID-19 into a term for SARS-CoV-2 (or the reverse). For example, adding "associated coronavirus" to the end of any term for COVID-19 results in a term for SARS-CoV-2. Likewise, adding "infection" to any term for SARS-CoV-2 results in a term for COVID-19. The third type of rule expands an existing term for COVID-19 or SARS-CoV-2 into a term for the same entity. For example, adding "disease" to a term for COVID-19 results in another term for COVID-19. The rules are applied repeatedly, resulting in 101,206 potential terms for COVID-19 and 413 potential terms for SARS-CoV-2.

The next step in our approach is to identify instances of the potential terms in the biomedical literature. Our dictionary tagger identifies terms independent of case and without regard for whitespace or punctuation. The dictionary tagger initially identifies all matching terms, so that "COVID-19 disease" results in three mentions: "COVID," "COVID-19," and "COVID-19 disease." However, embedded mentions are dropped, in this case leaving only "COVID-19 disease." Terms that do not appear in the results at least once are filtered.

The final step in our approach is manual review of all terms found, including a sample of appearances in context to ensure their usage in context is not ambiguous. We identify rules that should be added, such as the near-synonym relationship between "associated" and "related." These steps are iterated until additional rules cannot be identified.

# 3   Results

We apply our dictionary to LitCovid (Chen et al., 2020), a curated literature hub of PubMed articles on COVID-19 that is updated daily. LitCovid is the largest collection of articles specific to COVID-19, containing 43,448 documents (6,516,832 tokens) as of August 27, 2020. Our dictionary tagger, described in Section 2.2, identified 424 unique terms for COVID-19 from 116,577 mentions; of these 92,395 (79.3%) are equivalent to the mention "COVID-19" after normalizing case and removing whitespace and punctuation. The tagger also identified 126 unique terms for SARS-CoV-2 from 31,732 mentions; of these 22,559 (71.1%) are equivalent to the mention "SARS-CoV-2" after normalizing case and removing whitespace and punctuation. Due to the lack of gold-standard corpora containing mention-level annotations of COVID-19 and SARS-CoV2, we do not provide precision/recall evaluation for our dictionary.

## 3.1   Comparison

We compare our dictionary of terms for COVID-19 and SARS-CoV-2 against the terms in two other resources. The first dictionary, *AitsLabCorona*, is derived from the COVID-19 and SARS-CoV-2 dictionaries from the Aits Lab Corona project (Rashed et al., 2020) by manually assigning terms that are listed for both COVID-19 and SARS-CoV-2 to only one entity. The second dictionary, *Combined*, is a collection of terms for COVID-19 and SARS-CoV-2 from WikiData (Waagmeester et al., 2020) and eight biomedical terminologies: NCI Thesaurus (Fragoso et al., 2004), Monarch Disease Ontology (MONDO) (Shefchek et al., 2020), Medical Subject Headings (MeSH), Unified Medical Language System (UMLS) (Bodenreider, 2004), NCBI Taxonomy (Federhen, 2012), SNOMED CT[4], Disease Ontology (Kibbe et al., 2015), and FDA Substance Registration System (UNII)[5]. Terms listed for both COVID-19 and SARS-CoV-2 are again manually assigned to only one entity. Because our focus is terms that are used in the literature, we identify instances of the terms in the biomedical literature, normalizing case and removing both punctuation and whitespace as we do for our dictionary. Terms that do not appear in the results at least once are filtered.

---

[4] https://www.snomed.org/

[5] https://fdasis.nlm.nih.gov/srs/

We compare the terms found in each COVID-19 and SARS-CoV-2 term dictionary by identifying the terms that appear in the each of the dictionaries. The results are presented as a Venn diagram in Figure 2. Analysis of the comparison shows our dictionary includes many single-token variations the others lack, such as "SARS-CoV-2 infected" and "2019-new CoV." However, our dictionary lacks terms that include "pandemic" and is missing semantically inverted terms such as "disease caused by SARS-CoV-2." The *Combined* dictionary also contains the term "coronavirus" as a synonym for SARS-CoV-2[6]. This term does not appear in our dictionary because the sense of "coronavirus" referring specifically to SARS-CoV-2 is uncommon in the scientific literature.

## 4    Discussion

While the number of unique terms for SARS-CoV-2 and COVID-19 is large, a single term is responsible for most mentions of each concept. This suggests that an extensive dictionary may not be necessary. To determine if this is the case, we analyzed the frequency that the terms in our dictionary appear in LitCovid. The distribution is long tail for both entities: the average frequency of terms for COVID-19 is 275 with a median of 2, while the average frequency of terms for SARS-CoV-2 is 252 with a median of 3. Plotting the frequency of each term against its rank on a log-log scale confirms that both distributions are Zipfian (data not shown).

Since most terms are uncommon, however, the additional recall provided by each additional term is small. Thus, achieving high recall does require an extensive dictionary. For example, identifying 99% of the COVID-19 mentions identified by our dictionary requires 45 terms (27 for SARS-CoV-2) but identifying 99.5% of the COVID-19 mentions requires 148 terms (89 for SARS-CoV-2). Note that these values are likely underestimated, since our dictionary likely does not contain all terms for either entity. Moreover, while some of the uncommon terms (e.g. "COVID-2019 pneumonia") contain a more common term (e.g. "COVID-2019"), many variations are not simple extensions: "COVID-2019" does not contain "COVID-19". Thus, even for tasks where a term could be considered redundant if it contains a more common term, the list of non-redundant terms

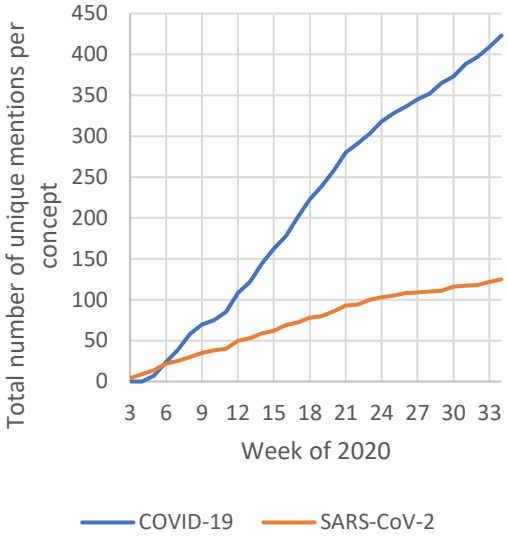

Figure 3: Total number of unique terms for COVID-19 and SARS-CoV-2 over time

remains relatively long (112 for COVID-19, 39 for SARS-CoV-2).

### 4.1    Term usage over time

We analyzed the terminology used in the literature to refer to COVID-19 and SARS-CoV-2 over time. We used the dictionary tagger results on LitCovid with our dictionary of terms for COVID-19 and SARS-CoV-2. We represented time as the week that the article was initially added to PubMed, which is typically earlier than the publication date.

Figure 3 shows the total number of unique terms used in the literature to refer to COVID-19 or SARS-CoV-2 as a function of time. The standard names for COVID-19 and SARS-CoV-2 were proposed in week 7, but the trends for both entities after that time remain highly linear. This data suggests both that a high degree of term variation must be handled to comprehensively identify all terms for either entity, and the dictionary will require regular updates.

In contrast, however, Figure 4 shows the percentage of articles (per week) that reference either entity using the most common mention for the entity ("@1") or one of the 5 most common terms for the entity ("@5"). The 5 most common terms for COVID-19 are, in order of descending frequency: "COVID-19," "SARS-CoV-2 infection," "Coronavirus disease 2019," "COVID," and "Novel coronavirus pneumonia." The five most common terms for SARS-CoV-2,

---

[6] From https://www.wikidata.org/wiki/Q82069695

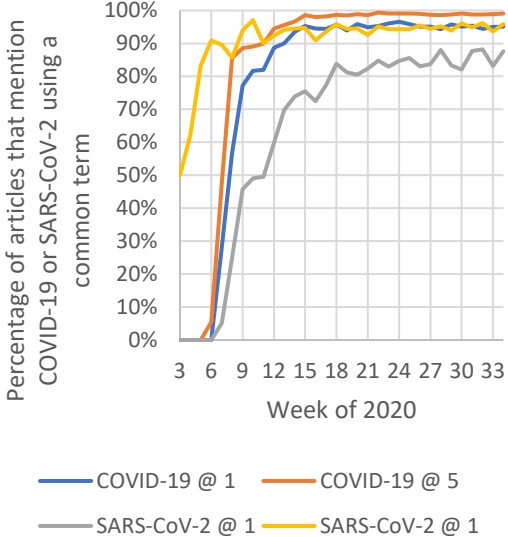

Figure 4. Percentage of articles that mention COVID-19 or SARS-CoV-2 using a common term.

again in order of descending frequency, are "SARS-CoV-2," "Novel coronavirus," "Severe acute respiratory syndrome coronavirus 2," "2019-nCov," and "2019 novel coronavirus." This figure demonstrates that despite the high degree of term variation, most terms used are common, suggesting that relatively high automated performance can be obtained using relatively common terms. The figure also shows that the term "SARS-CoV-2" was accepted more slowly than "COVID-19," and that the usage rate for the term "SARS-CoV-2" remains lower than the rate for "COVID-19." Finally, the continued appearance of the term "novel coronavirus" in the 5 most common terms is concerning because it appears in the literature hundreds of times before the pandemic, making its meaning both ambiguous and dependent on the temporal context.

## 5    Conclusion

We presented a dictionary of terms used to refer to COVID-19 and SARS-CoV-2 in the biomedical literature. Our resource cleanly separates terms for COVID-19 and terms for SARS-CoV-2 and our comparison demonstrates that it provides a substantial number of additional synonyms not available in the existing terminologies of which we are aware. We will regularly add new terms and our dictionary is freely available at https://github.com/ncbi-nlp/CovidTermVar.

Substantial limitations of our approach include the need to manually verify results and the limited

domain, namely two specific entities. Other studies have previously identified near synonyms as a significant source of synonymy for biomedical terms (Blair et al., 2014), and generalizing this approach to address normalization of a wider range of entities may be potentially interesting. There is also a significant amount of prior work on identifying terminology with reduced supervision (Neelakantan & Collins, 2014; Riloff & Shepherd, 1999; Williams et al., 2015), which would reduce manual involvement and/or allow more entities to be considered. Finally, it would be valuable to automatically identify the most important terms to review systematically (Tsuruoka et al., 2008).

## Acknowledgments

This research was supported by the Intramural Research Program of the National Library of Medicine at the NIH.

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
