# OpenReview forum: "A Comprehensive Dictionary and Term Variation Analysis for COVID-19 and SARS-CoV-2"
_EMNLP/2020/Workshop/NLP-COVID — NLP-COVID19-EMNLP Poster_

### Official Review · AnonReviewer2 · 2020-09-17
**Interesting paper but with limited applicability**

**Rating:** 4
**Confidence:** 4

**Review:**

The paper describes the generation of a dictionary of terms related to COVID-19 based on a set of rules. The rules allows to create a set of different combinations using a set of base-terms that allows to identify mentions to the disease by detecting terms that are not very common. The results showed that the approach developed by the authors can have interesting applications for example in bibliometrics as the detection of those concepts less used can help in this field. From the NLP perspective, the technical advances are very limited although the results provided are reasonable useful.

---

### Official Review · AnonReviewer3 · 2020-09-24
**A good description of a very useful push request**

**Rating:** 4
**Confidence:** 4

**Review:**

This paper present a technique to expand gazetteers referring to Covid19 (disease) and Sars-cov-2 (virus), consisting of a set of rules.
In addition, there is an analysis of the result applying it to a collection called LitCovid.


My biggest concern with this paper is that it is presented as a way of expanding coverage of automated tools. However, that utility is never shown. This by itself might not be an issue with a workshop paper. However, there are reasons to believe that it will be limited: the novel terms belong probably to the long tail and are very rarely used, and maybe even in a redundant way with the more standard terms.
In that sense, Table 1 provides the wrong information. It is not the number of terms that is important, but the combined number of occurrences of those terms in a given document. As ~80% (70%) of the terms referring to Covid19 (Sara-cov-2) are covered by the basic expression, it would seem that the percentage that is covered by your expanded list might be very small. Fig. 3 addresses this with the top-1/2/5 terms, but it is not clear how many additional terms (or even documents) are captured with your expanded list.

There could be value in analyzing in what situations non-standard terms are used, and if those are preferred by a certain public or region. However, besides Fig 1 (which I found one of the most interesting parts of this paper), such an analysis is absent.

This list would be very helpful, and will hopefully be provided as a push-request to the terminology list referred in the paper (https://github.com/Aitslab/corona)

Other comment:
  - What is LitCovid: in order for the paper to be self-contained, specifying the type, source and some statistics on the collection would be helpful

---

### Official Review · AnonReviewer4 · 2020-09-27
**Useful resource, highly targeted**

**Rating:** 6
**Confidence:** 4

**Review:**

The authors have developed rule-based strategies to generate term variants for the SARS-CoV-2 virus and COVID-19 disease and examined the usage of these variants in the literature.

They distinguish the task from NER; I agree it is not NER, but I do think it is comparable to normalization to a very small set of (two) identifiers. In that light, some assessment of the quality of the terminology in terms of Precision/Recall would potentially be valuable. Are any of the terms ambiguous? The authors also assume that if a term is instantiated in the literature, it is valid. Is it possible that there is a spurious match between a term and the literature? Furthermore, given the comment about the need for regular updates, why not just leave the terms in the dictionary? Perhaps those terms can anticipate variation?

I'm a little bit confused by the analysis in Table 1. If the Union of all of the resources is considered the "complete" dictionary, shouldn't we be interested in how many terms are unique to each resource? That is, how many terms are added from each source that contribute to the final set of terms? And, shouldn't we consider the term-level overlap between them in order to establish agreement, e.g. which terms are in all 3, 2, or only a single resource?

Figure 2 could be more interesting if we were also told the proportion of articles each of these unique terms appeared in, because the literature was also increasing during that time; this would give a sense of the rate of adoption of these 'novel' terms. Perhaps some 'error bars' could suggest the variation in the distribution of the terms.

The authors in Section 2.1 identify 3 types of entities: The SARS-CoV-2 virus, the disease it causes (COVID-19), and the pandemic. So the third type (naming the pandemic) is excluded from the dictionary?

I'm wondering how generalizable/productive the rules are that expand the terms. Some are obviously highly specific to this virus, but others such as adding "infection" to the name of a virus to produce the disease, or adding "disease" to the end of a base disease term, seem like they would be generalizable. It would be interesting to see whether such patterns apply for other virus-disease associations.

All in all, a useful resource in a narrow context. It would be nice to understand the broader utility of the methodology.